# Deciphering AKI in Burn Patients: Correlations between Clinical Clusters and Biomarkers

**DOI:** 10.3390/ijms25126769

**Published:** 2024-06-20

**Authors:** Shin Ae Lee, Dohern Kym, Jaechul Yoon, Yong Suk Cho, Jun Hur, Dogeon Yoon

**Affiliations:** 1Department of Surgery and Critical Care, Burn Center, Hangang Sacred Heart Hospital, Hallym University Medical Center, 12, Beodeunaru-ro 7-gil, Youngdeungpo-gu, Seoul 07247, Republic of Korea; golove4044@gmail.com (S.A.L.); justinoj@hallym.or.kr (J.Y.); maurchigs@hallym.or.kr (Y.S.C.); hammerj@hallym.or.kr (J.H.); 2Burn Institutes, Hangang Sacred Heart Hospital, Hallym University Medical Center, 12, Beodeunaru-ro 7-gil, Youngdeungpo-gu, Seoul 07247, Republic of Korea; hyeonyoon@hallym.or.kr

**Keywords:** AKI, burns, cluster analysis, longitudinal study, decision curve analysis

## Abstract

Acute kidney injury (AKI) is a significant complication in burn patients, impacting outcomes substantially. This study explores the heterogeneity of AKI in burn patients by analyzing creatinine time-series data to identify distinct AKI clusters and evaluating routine biomarkers’ predictive values. A retrospective cohort analysis was performed on 2608 adult burn patients admitted to Hangang Sacred Heart Hospital’s Burn Intensive Care Unit (BICU) from July 2010 to December 2022. Patients were divided into four clusters based on creatinine trajectories, ranging from high-risk, severe cases to lower-risk, short-term care cases. Cluster A, characterized by high-risk, severe cases, showed the highest mortality and severity, with significant predictors being PT and TB. Cluster B, representing intermediate recovery cases, highlighted PT and albumin as useful predictors. Cluster C, a low-risk, high-resilience group, demonstrated predictive values for cystatin C and eGFR cys. Cluster D, comprising lower-risk, short-term care patients, indicated the importance of PT and lactate. Key biomarkers, including albumin, prothrombin time (PT), cystatin C, eGFR cys, and total bilirubin (TB), were identified as significant predictors of AKI development, varying across clusters. Diagnostic accuracy was assessed using area under the curve (AUC) metrics, reclassification metrics (NRI and IDI), and decision curve analysis. Cystatin C and eGFR cys consistently provided significant predictive value over creatinine, with AUC values significantly higher (*p* < 0.05) in each cluster. This study highlights the need for a tailored, biomarker-driven approach to AKI management in burn patients, advocating for the integration of diverse biomarkers in clinical practice to facilitate personalized treatment strategies. Future research should validate these biomarkers prospectively to confirm their clinical utility.

## 1. Introduction

Acute kidney injury (AKI) in burn patients represents a significant clinical challenge, with the critical nature of burn injuries necessitating a deeper understanding of this complication. Burns, being one of the most severe forms of trauma, cause both localized damage and systemic implications, dramatically affecting patient outcomes [1]. The pathophysiological changes following severe burns can lead to significant morbidity and mortality, making effective management crucial [2]. AKI, a common complication in burn patients, presents additional clinical challenges. Defined by a rapid decline in renal function, AKI is associated with increased mortality rates and prolonged hospital stays [3]. The prevalence of AKI in burn patients and its impact on outcomes underscores the importance of understanding this complication in the context of burn injuries [4].

The complexity of AKI, particularly in burn injuries, is compounded by its heterogeneity. AKI is not a uniform syndrome; instead, it manifests variably across different patient populations, influenced by a range of factors including etiology and underlying health conditions [5]. In burn patients, the size, depth, and associated trauma of the burn significantly influence the development and progression of AKI [6]. The variability in AKI’s presentation necessitates a comprehensive study to understand its many manifestations and impacts on patient outcomes [7]. This variability also highlights the severity and heterogeneity of burns themselves and their relevance to AKI, emphasizing the importance of research in critical and heterogeneous diseases.

Recognizing this heterogeneity, this study supports a detailed subgroup analysis to uncover variations in AKI presentation among burn patients. A subgroup analysis in clinical research identifies patterns and subtypes within a diverse patient population, providing valuable insights for tailored treatment strategies [8]. Such an approach is particularly relevant in the era of precision medicine, where individual patient characteristics and disease subtypes are increasingly guiding treatment decisions [9].

A distinguishing feature of this study is the utilization of longitudinal data to identify AKI subgroups in burn patients. Unlike cross-sectional studies, longitudinal studies track patients over time, providing a dynamic view of disease progression and potential predictive markers [10]. This approach is crucial for understanding the temporal patterns of AKI in burn patients, including the significance of creatinine in the diagnosis and prognosis of AKI and its time-series changes for clustering in research.

The primary aim of this study is to elucidate the heterogeneity of AKI in burn patients by using creatinine time-series changes to identify distinct clusters. This involves identifying routine biomarkers that are predictive of AKI diagnosis across these clusters. By doing so, the study aims to facilitate early identification of subgroups at risk of developing AKI, thereby informing targeted treatment strategies and improving patient outcomes in this critical and heterogeneous patient population.

## 2. Results

### 2.1. Demographics for Overall Patients

The study included 2654 patients admitted to the ICU with burn injuries. After accounting for those without a complete creatinine trajectory (17 patients) and those missing initial predictors (29 patients), the final analysis comprised 2608 patients. These patients were categorized into four clusters based on the time-series clustering analysis. Cluster A included 461 patients, Cluster B had 730, Cluster C encompassed 453, and Cluster D, the largest, contained 964 patients (Figure 1).

The cohort was characterized by a majority of male patients, a mean age in the early fifties, and a predominance of flame burns. Inhalation injuries were also common. The mean hospital stays and total body surface area (TBSA) affected were indicative of the severity of injuries. A notable proportion of patients experienced sepsis, ARDS, and AKI, and many required interventions such as mechanical ventilation and continuous renal replacement therapy (CRRT). The observed mortality rate was significant, reflecting the seriousness of burn injuries. The study also reported average scores for APACHE IV and SOFA, underscoring the acute health risks to the patient group (Table 1).

### 2.2. Characteristics of Each Cluster

In our study, we delineated four patient subgroups through time-series clustering, each with distinct clinical trajectories and healthcare need (Figure 2, Table 2).

Cluster A, which we have designated as the ‘high-risk and high-severity subgroup’, is characterized by the highest mortality rate at 48.6% and the longest hospital stays, averaging 21.5 days. This cluster typically includes older patients, averaging 55.7 years old, with extensive injuries as indicated by a higher mean total body surface area (TBSA) affected. Clinically, this group exhibits the most severe outcomes, as reflected by the highest scores on the Hangang, APACHE IV, and SOFA scales. The clinical implication for this subgroup suggests a need for intensive care and monitoring, as evidenced by the high incidence of sepsis, ARDS, and the frequent requirement of ventilators and CRRT.

The ‘intermediate-risk and recovery subgroup’, or Cluster B, presents with moderate mortality rates and intermediate lengths of hospital stay, about 23.6 days on average. Patients in this subgroup tend to have middle-range TBSA involvement and exhibit intermediate clinical severity scores. The clinical significance here is a potential for recovery, but with a need for substantial medical intervention and possible rehabilitation services.

Cluster C is recognized as the ‘low-risk and high-resilience subgroup’, with a lower mortality rate of 34.1% and the shortest average stay in the hospital, suggesting a more resilient patient profile. This cluster’s patients have the least severe clinical scores and the lowest incidence of sepsis and ARDS, which may indicate a better prognosis and a lower need for invasive treatments.

Lastly, Cluster D, termed the ‘low-risk and short-term care subgroup’, shares a low mortality rate similar to Cluster B but with shorter hospital stays, hinting at a need for less intensive but prompt medical care. The youngest patients fall into this cluster, and they exhibit the lowest clinical severity, suggesting that while their overall risk is lower, they require efficient and effective acute care to prevent complications. Patient trajectories for creatinine levels are depicted in Appendix A, illustrating the progression over time. Additionally, the variation in creatinine levels with hospital day is systematically tabulated in Appendix A.

### 2.3. Multivariable Logistic Analysis and Diagnostic Accuracy

Our investigation implemented a multivariable logistic regression analysis to ascertain the odds ratios of predictors for AKI, followed by an assessment of their diagnostic accuracy using AUC (Table 3). Significant predictors were identified based on their adjusted odds ratios, reflecting the strength of association with AKI incidence after adjusting for confounding factors (Figure 3 and Appendix A).

In Cluster A, significant predictors determined through logistic regression include albumin, PT, RDW, creatinine, cystatin C, eGFR cys, and TB, with AUC values ranked as follows: PT with 0.918, cystatin C with 0.917, eGFR cys with 0.908, albumin with 0.892, TB with 0.887, RDW with 0.886, and creatinine with 0.885.

For Cluster B, albumin, PT, creatinine, and WBC emerged as significant, exhibiting AUCs of 0.875 for albumin, 0.874 for creatinine, 0.871 for PT, and 0.848 for WBC.

Within Cluster C, the logistic regression identified albumin, creatinine, eGFR cr, platelet, PT, pH, BUN, cystatin C, TB, lactate, RDW, and eGFR cys as significant predictors. The highest AUC was for Alb at 0.840, followed by PT at 0.804, cystatin C at 0.791, eGFR cys at 0.780, creatinine at 0.764, and pH at 0.764.

Lastly, Cluster D’s significant predictors were albumin, PT, platelet, lactate, RDW, creatinine, cystatin C, eGFR cys, WBC, and TB, with AUC values leading with albumin at 0.798, PT at 0.757, lactate at 0.744, cystatin C at 0.731, and platelet at 0.729.

### 2.4. Significant Predictors Using Reclassification Metrics by Cluster

In Cluster A, cystatin C (IDI: 0.173, NRI: 0.854), PT (IDI: 0.136, NRI: 0.764), and eGFR cys (IDI: 0.141, NRI: 0.533) outperformed creatinine in both IDI and NRI. TB also showed better performance with an IDI of 0.035 and NRI of 0.374. Albumin, although it did not surpass the reference in IDI, was noteworthy in NRI with a value of 0.401.

In Cluster B, PT and albumin were noted for their significance in NRI (PT NRI: 0.261; albumin NRI: 0.169) without corresponding significance in IDI.

For Cluster C, albumin stood out significantly in both IDI (0.125) and NRI (0.717), as did PT with IDI of 0.049 and NRI of 0.614. Cystatin C was also a strong predictor, with positive and significant values in both IDI (0.028) and NRI (0.256). Lactate and eGFR cys were acknowledged for their positive NRI value of 0.202 and 0.048, respectively.

In Cluster D, albumin (IDI: 0.095, NRI: 0.764), PT (IDI: 0.029, NRI: 0.439), and cystatin C (IDI: 0.019, NRI: 0.188) were identified as significant in both IDI and NRI, whereas eGFR cys and lactate had a notable NRI only.

Figure 4 and Figure 5, along with Appendix A, present our findings on biomarkers that offer statistically significant improvements in predicting AKI over creatinine. These biomarkers enhance the classification and differentiation of patient risk, promising to refine AKI prognostic models and aid in clinical decision-making across patient clusters.

### 2.5. Assessing Biomarker Efficacy in AKI: Insights from Decision Curve Analysis

The decision curve analysis, presented in Figure 6 and elaborated in Appendix A, was conducted to evaluate the net benefit of various biomarkers in predicting acute kidney injury (AKI). This study specifically honed in on biomarkers that were pivotal in either or both IDI and NRI assessments.

Within Cluster A, the analysis pinpointed cystatin C, PT, eGFR cys, albumin, and TB as significant in both NRI and IDI evaluations. These biomarkers demonstrated superior net benefit compared to creatinine across a spectrum of threshold probabilities. Notably, cystatin C and eGFR cys outshined creatinine consistently above the 30% threshold probability. PT and TB also exhibited greater net benefit in nearly all measured intervals, while albumin showed enhanced benefit in select intervals. These results underscore the potential of cystatin C, eGFR cys, PT, and TB as more reliable indicators for directing AKI treatment strategies at critical clinical junctures.

The findings in Cluster B indicated a limited predictive capacity for creatinine, showing negative net benefits at certain intervals. This cluster lacked significant predictors in both NRI and IDI assessments. However, cystatin C and eGFR cys still demonstrated a greater net benefit than creatinine, except in the 95% threshold probability.

In Cluster C, albumin, PT, cystatin C, lactate, and eGFR cys emerged as significant predictors in either NRI or IDI assessments. Yet, the competitive nature of these biomarkers was evident, as they showed similar net benefits to creatinine across most threshold probability.

The analysis of Cluster D revealed that PT, cystatin C, and eGFR cys offered net benefits comparable to creatinine for most threshold probabilities. This consistency highlights the nuanced role these biomarkers play in the predictive landscape of AKI.

The decision curve analysis underscores the need for a cluster-specific evaluation of biomarkers in the context of AKI. It reveals that the effectiveness of additional biomarkers over creatinine can vary depending on patient demographics and clinical characteristics, highlighting the importance of personalized approaches in medical decision-making.

## 3. Discussions

This study intricately dissects acute kidney injury (AKI) in burn patients, employing a detailed subgroup analysis that highlights the heterogeneity of AKI and its correlation with clinical factors such as cystatin C, eGFR cys, albumin, PT, TB, and lactate. By categorizing patients into distinct clusters—from the ‘high-risk and high-severity subgroup’ to the ‘low-risk and short-term care subgroup’—it illuminates the varying degrees of clinical urgency, recovery potential, and resilience. The study’s meticulous approach and integration of longitudinal data offer profound insights into the temporal dynamics of AKI in burn patients, advocating for a more personalized, biomarker-driven approach to AKI management. However, translating these insights into clinical practice requires cautious optimism, rigorous validation, and an understanding of the study’s limitations.

Notably, cystatin C and eGFR cys were consistently prominent across all clusters, showing significant potential as predictors for AKI treatment decisions. In Cluster A, PT and TB also emerged as significant, particularly for high-risk, severe burn patients. Meanwhile, PT and albumin showed predictive potential in Cluster B, with PT and lactate being noteworthy in Cluster D, and Cluster C demonstrating similar benefits. This analysis confirms the central role of cystatin C and eGFR cys in all subgroups, while also highlighting the relevance of albumin, PT, TB, lactate, and creatinine in specific clusters. 

To provide a holistic view, it is essential to compare and integrate the findings from all clusters. Cluster A, which comprises high-risk patients, shows the highest mortality and severity, suggesting the need for intensive monitoring and treatment. The predictive value of PT and TB in this cluster underscores the severity of systemic involvement in these patients. In contrast, Cluster B, characterized by intermediate recovery, indicates that PT and albumin are useful predictors, reflecting a different clinical trajectory where recovery is plausible but still requires substantial medical intervention. Cluster C, the low-risk, high-resilience subgroup, shows the lowest severity and the shortest hospital stays, where predictors like cystatin C and eGFR cys still play a crucial role but suggest a generally better prognosis. Cluster D, another low-risk group with short-term care needs, highlights the importance of PT and lactate, suggesting that while these patients are at lower risk, they still benefit from close monitoring to prevent complications. 

This integrated analysis of the clusters reveals the nuanced roles that different biomarkers play across varying clinical contexts, advocating for a targeted and patient-focused approach in biomarker selection for AKI prediction. Understanding the interactions and comparative outcomes across these clusters allows for more precise and individualized treatment strategies, ultimately improving patient care outcomes. While our study highlights the predictive power of cystatin C and eGFR cys, it is important to acknowledge that these findings require further validation in larger, multicenter cohorts to ensure their generalizability and robustness across different clinical settings.

Cystatin C, in particular, has garnered attention for its potential in refining the estimation of glomerular filtration rate (eGFR) when traditional creatinine-based estimates fall short. Its ability to provide a more precise assessment of renal function is crucial, especially in the context of burn patients, where the accurate and timely diagnosis of AKI is vital for effective management. Toffaletti [11] underscored the clinical utility of cystatin C, noting its increasing adoption in routine practice and its strong correlation with true GFR measurements. This body of evidence corroborates the findings of the current study, particularly in high-risk clusters where the precision of AKI diagnosis and management is paramount. However, while the utility of cystatin C in enhancing the precision of GFR estimates is acknowledged, studies suggest the need for cautious implementation, especially considering cost-effectiveness in primary care settings and accuracy in specialized populations [12,13].

Prothrombin time (PT) and total bilirubin (TB) are additional biomarkers highlighted in this study, primarily associated with liver function and overall patient health status. Elevated levels of PT and TB often indicate liver dysfunction, a condition that can have significant implications for renal function and AKI risk. The association between liver dysfunction markers and AKI risk, noted in various studies, supports this study’s findings and emphasizes the interconnected nature of organ systems in critically ill patients, such as those with severe burns [14,15].

Albumin, another biomarker emphasized in this study, is widely recognized for its diagnostic and prognostic value in various clinical settings. Its levels can reflect the nutritional status, inflammation, and overall health of the patient, factors that are particularly pertinent in the management of burn patients, where nutritional support and inflammation control are crucial [16]. The association between hypoalbuminemia and AKI development, seen in various studies, resonates with the findings of this study, underscoring the multifaceted role of albumin in patient assessment and management [17,18].

Lactate, in particular, has been recognized as an important indicator of tissue hypoperfusion and a predictor of AKI development. Research has shown that elevated serum lactate levels, often seen in critically ill patients such as those with traumatic brain injury, are indicative of systemic tissue hypoperfusion due to factors like massive blood loss, cardiopulmonary dysfunction, or an activated renin–angiotensin system. This hypoperfusion can lead to renal tissue hypoxia, a condition strongly associated with the development and progression of AKI. The study’s findings on lactate align with the existing literature, which supports its role as a prognostic marker in critically ill patient populations, reflecting the systemic impact of severe conditions like burns and its correlation with AKI risk [19,20,21].

In clinical practice, the application of these biomarkers for AKI management is of paramount importance. The integration of biomarker analysis into clinical decision-making allows for a more refined approach to patient care. By identifying patient subgroups at higher risk for AKI, clinicians can prioritize resources, tailor treatment protocols, and potentially initiate preventive strategies. Moreover, monitoring these biomarkers can aid in tracking treatment efficacy and adjusting interventions in real time, thereby optimizing patient outcomes.

Despite the promising implications of this study, it is imperative to acknowledge its limitations. Our study’s focus on a single-center cohort limits the generalizability of the findings to other settings. The significant variation in mortality rates, length of stay, and other clinical parameters within clusters suggests heterogeneity that may lead to overlapping characteristics and less precise clinical recommendations. Additionally, the ten-year span of the cohort introduces variability due to evolving clinical practices, treatment protocols, and medical technologies. This temporal variability complicates the attribution of results solely to patient characteristics or interventions studied. The 2–3-day interval for creatinine measurements may have missed some acute changes, potentially impacting the accuracy of AKI diagnosis and timing. Furthermore, the retrospective design and the reliance on historical data may introduce biases, and the lack of a prospective validation cohort precludes the immediate clinical application of the study’s conclusions.

## 4. Methods

### 4.1. Study Design and Patient Selection

We conducted a retrospective cohort study at Hangang Sacred Heart Hospital’s Burn Intensive Care Unit (BICU) from July 2010 to December 2022. The study included 2654 adult burn patients aged 18 and older. Exclusions were made for patients under 18, those without burn injuries, patients lacking creatinine data for more than two days, and those missing all admission predictors. The study was conducted in adherence to the STROBE guidelines.

### 4.2. Data Collection and Outcome Measures

Data were retrieved from a prospectively maintained data lake at Hallym University Medical Center, encompassing a wide range of parameters including demographics, diagnoses, and ICU stay durations. At the time of BICU admission, the study collected the following specific predictors: minimum pH, prothrombin time (PT), albumin, creatinine (Cr), blood urea nitrogen (BUN), total bilirubin (TB), white blood cell count (WBC), lactate, platelet count, red cell distribution width (RDW), cystatin C, creatinine GFR, and cystatin GFR. Additionally, severity scores such as APACHE IV, SOFA score, ABSI, rBaux, and Hangang score [22] were also gathered. All biomarkers were measured from blood samples collected from patients at the time of ICU admission and during their stay. Our primary focus was on assessing the incidence of acute kidney injury (AKI), with a particular emphasis on the 60-day in-hospital mortality. In the ICU, creatinine levels were monitored from admission to discharge, with measurements taken at least every 2–3 days.

AKI diagnosis and severity classification were based on the AKIN criteria. Stage 1 includes a serum creatinine increase of ≥0.3 mg/dL (≥26.5 µmol/L) within 48 h, 1.5–1.9 times the baseline in the past 7 days, or urine output less than 0.5 mL/kg/h for 6–12 h. Stage 2 is characterized by a serum creatinine increase to 2.0–2.9 times the baseline or urine output less than 0.5 mL/kg/h for over 12 h. Stage 3 involves a serum creatinine increase to 3 times the baseline, ≥4.0 mg/dL (≥353.6 µmol/L), initiating renal replacement therapy, a decrease in eGFR to less than 35 mL/min/1.73 m^2^ in patients under 18, or urine output less than 0.3 mL/kg/h for over 24 h or anuria for over 12 h. The lowest value during hospital admission was used as the baseline serum creatinine, or it was estimated using the MDRD equation assuming a baseline eGFR of 75 mL/min/1.73 m^2^. The CRRTs at our burn center were acidosis, azotemia, electrolyte imbalance, volume overload, hypercreatininemia, hyperlactatemia, and the discretion of the attending burn surgeon. CRRT was performed using the continuous veno-venous hemodiafiltration mode, with a blood flow rate of 125 mL/min and an intensity of 2000 mL/h (dialysate: 1000 mL/h and replacement: 1000 mL/h). Hemosol BO fluid (Gambro, Lund, Sweden.) was used as the dialysate and replacement fluid. Sepsis diagnoses adhered to sepsis-3 criteria, using the SOFA score to identify acute changes with a score of 2 or higher. ARDS was diagnosed according to the Berlin definition, with onset identified within one week of known trauma or new/worsening respiratory symptoms. Due to the retrospective nature of the study, informed consent was waived, with approval from the Institute Review Board (IRB) of Hangang Sacred Heart Hospital (HG2024-003).

### 4.3. Data Management and Clustering

In managing patient data, our procedures strictly conformed to established clinical practice norms and relevant data protection laws. We prioritized patient confidentiality by anonymizing all identifiable information. Additionally, we diligently worked to guarantee the accuracy and consistency of our data, resolving any discrepancies through careful examination of the original documents. Outlier management was addressed through the application of Winsorization techniques. All predictors were subjected to scaling prior to analysis to ensure uniformity of measurement scales. 

We employed the tsclust function from the ‘dtwclust’ R package, focusing on clustering time-series data to identify distinct patient subgroups. Our approach involved setting the cluster range between 3 and 8, which was crucial for pinpointing clinically relevant groups. We used dynamic time warping (DTW) as our distance measure to effectively align data with varying time intervals, and dynamic barycenter averaging (DBA) for accurate centroid computation, crucial for handling temporal shifts. To ensure consistency and remove scale bias, we applied Z-score normalization to the data. The clustering process was iteratively executed up to 300 times to guarantee the stability of our results, a key factor in the robust identification of patient subgroups.

To determine the optimal number of clusters for AKI subgroups, our analysis calculated the standard deviation of ‘Yes’ proportions for key conditions such as sepsis, ARDS, ventilator and CRRT use, and mortality across various clustering methods. We assessed the variability of these conditions within each cluster for different clustering configurations. The method with the highest count of conditions showing substantial differentiation was chosen based on a threshold of a standard deviation greater than 0.15. This approach ensured the identification of the most effective clustering method in capturing the heterogeneity in our study of acute kidney injury in burn patients.

### 4.4. Statistical Analysis

For normally distributed variables, we adopted the mean ± standard deviation approach, while non-normally distributed variables were described using the median and interquartile range (25th–75th percentile). Continuous variables were assessed using either the paired t-test or Wilcoxon signed-rank test, and categorical variables were examined using the Chi-squared test or Fisher’s exact test, depending on their distribution characteristics.

In the logistic analysis for AKI development, we included a range of factors like age, sex, TBSA, ARDS, sepsis onset, CRRT, and ventilator use, conducting a multivariable logistic analysis to identify significant predictors. This analysis was crucial for determining the adjusted odds ratios that helped in recognizing meaningful predictors for the development of AKI.

We evaluated the diagnostic accuracy of acute kidney injury (AKI) using a multivariable logistic regression model. Our assessment employed various metrics, including the area under the curve (AUC), which quantifies the model’s ability to distinguish between outcomes effectively. A high AUC indicates better model performance in differentiating cases with and without the condition. We also utilized the net reclassification improvement (NRI) and integrated discrimination improvement (IDI) measures. NRI helps in assessing how well the new predictors improve the risk classification of patients compared to the existing model, which is vital for understanding the clinical applicability of the model. IDI, on the other hand, measures the improvement in the model’s discriminatory power, showing how much the new predictors enhance the separation between those who develop the condition and those who do not.

Additionally, decision curve analysis was conducted to determine the clinical usefulness of the models, especially for variables significantly identified in IDI and NRI. This analysis helps in understanding the practical benefits of using these predictive models in a clinical setting, weighing the true positives against the false positives.

All these statistical analyses were conducted using the R software, version 4.3.0 (R Foundation for Statistical Computing, Vienna, Austria), ensuring a thorough and robust evaluation of the data to understand the intricacies of AKI in burn patients.

## 5. Conclusions

The study’s subgroup analysis and decision curve analysis provide a comprehensive understanding of the roles of albumin, PT, TB, and lactate as biomarkers in AKI management in burn patients. These biomarkers offer valuable insights into patients’ conditions, reflecting the complex interplay of various physiological processes in critical illnesses. The findings, supported by existing research, emphasize the importance of a multidimensional approach to patient assessment and management, integrating various biomarkers to capture the multifaceted nature of AKI in burn patients. By identifying specific biomarkers relevant to different patient subgroups, this study advocates for a personalized treatment strategy to enhance AKI management. Future research should aim to further explore the utility of these biomarkers in clinical practice. This should include prospective studies to validate their applicability and efficacy in improving patient care and outcomes. Ensuring that these biomarkers can be reliably used in diverse clinical settings will be crucial for their adoption in standard practice.

## Figures and Tables

**Figure 1 ijms-25-06769-f001:**
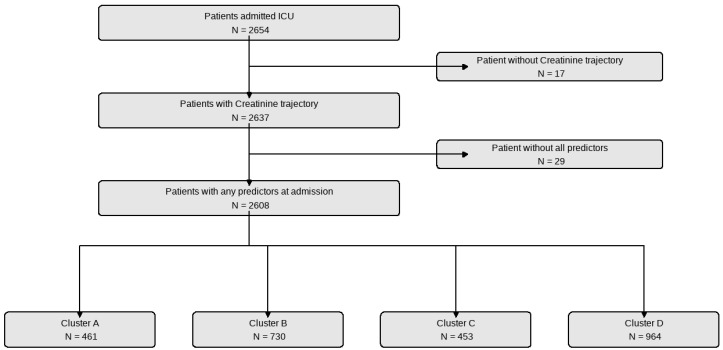
Patient selection and clustering process.

**Figure 2 ijms-25-06769-f002:**
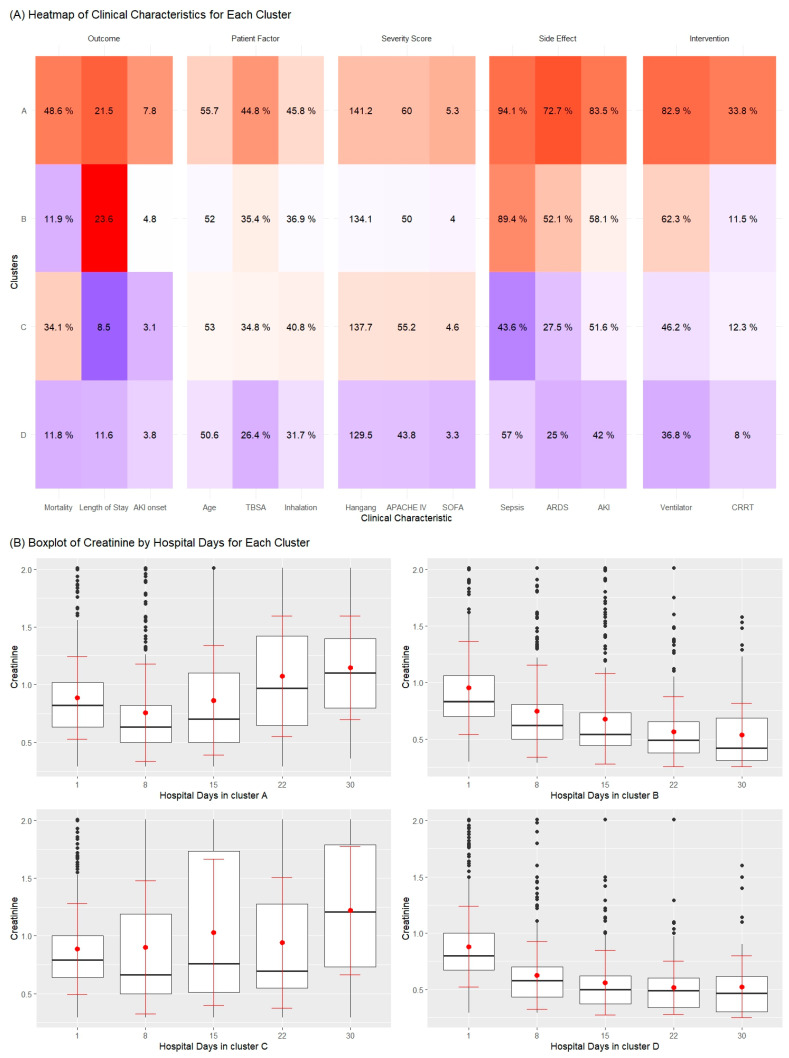
Heatmap (**A**) and boxplots (**B**) showcasing the distribution of clinical characteristics and creatinine levels, respectively, across patient clusters.

**Figure 3 ijms-25-06769-f003:**
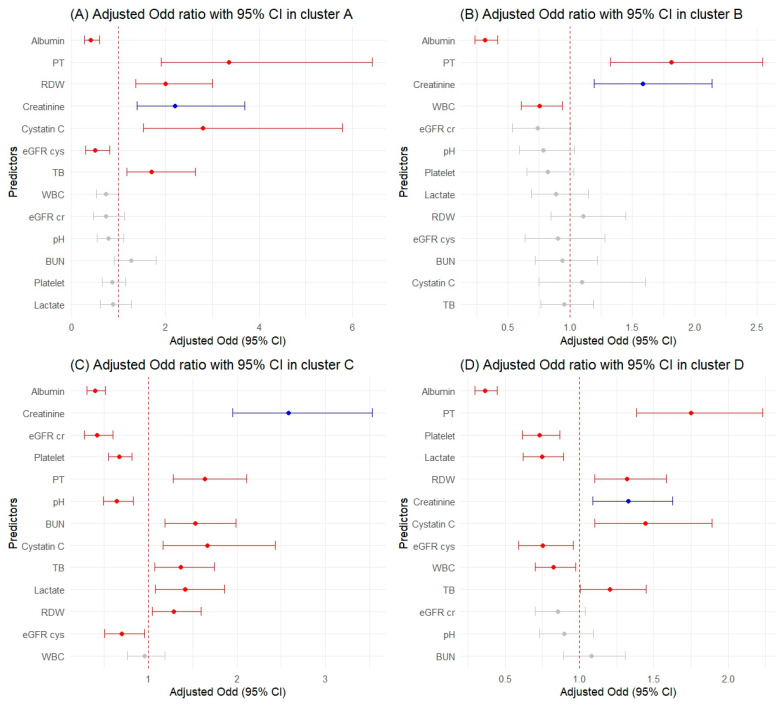
Adjusted odds ratios with 95% confidence intervals for selected predictors across clusters. The blue color represents the creatinine used in the trajectory. The red color indicates a *p*-value of 0.05 or lower, and the gray color signifies non-significant results.

**Figure 4 ijms-25-06769-f004:**
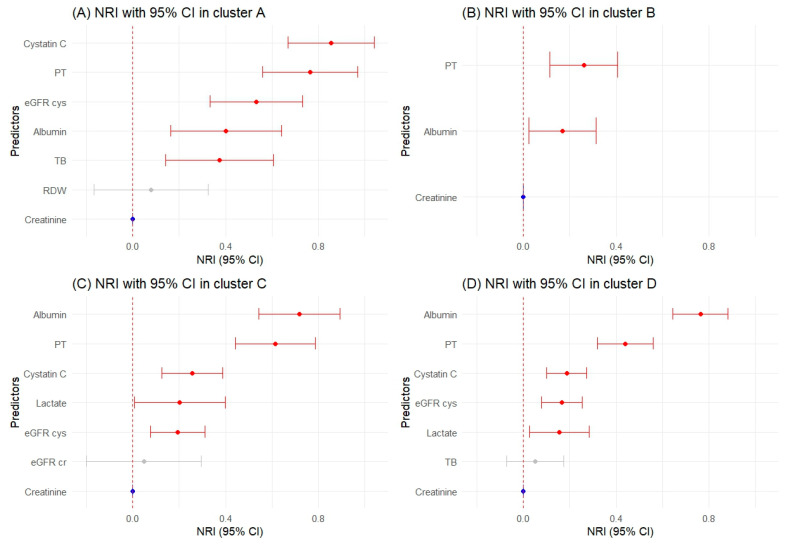
Net reclassification improvement (NRI) values with 95% confidence intervals. The blue color represents the creatinine used in the trajectory. The red color indicates a *p*-value of 0.05 or lower, and the gray color signifies non-significant results.

**Figure 5 ijms-25-06769-f005:**
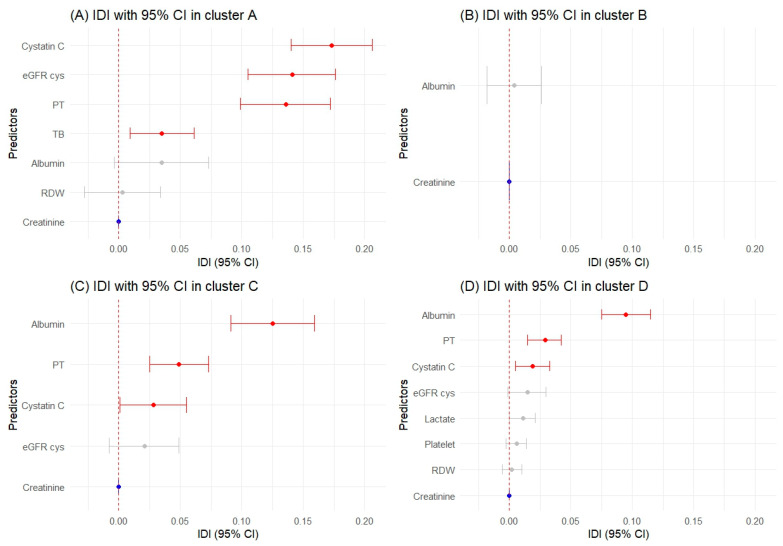
Integrated discrimination improvement (IDI) statistics across clusters. The blue color represents the creatinine used in the trajectory. The red color indicates a *p*-value of 0.05 or lower, and the gray color signifies non-significant results.

**Figure 6 ijms-25-06769-f006:**
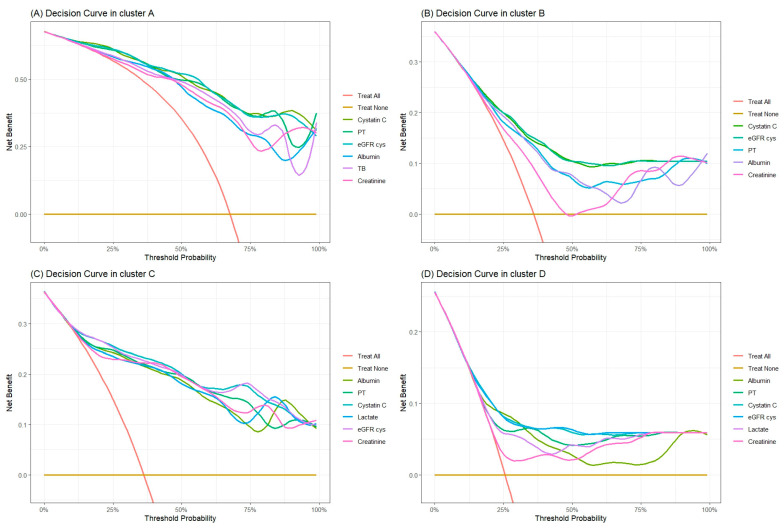
Decision curves highlight the clinical benefits of different predictive models based on patient clusters.

**Table 1 ijms-25-06769-t001:** Overall Demographic Characteristics for Enrolled Patients.

Group	Characteristic	N = 2608 ^1^
Outcomes	Mortality	641 (24.6%)
	Length of Stay	14.6 ± 10.7 (1.0–30.0)
	AKI onset	4.9 ± 6.4 (1.0–30.0)
Patients’ Factors	Age	52.4 ± 16.2 (18.0–99.0)
	Sex	
	Male	2067 (79.3%)
	Female	541 (20.7%)
	TBSA	33.5 ± 24.7 (0.0–100.0)
	Inhalation	982 (37.7%)
	Type	
	FB	1818 (69.7%)
	SB	257 (9.9%)
	EB	351 (13.5%)
	ChB	46 (1.8%)
	CoB	136 (5.2%)
Severity Scores	ABSI	8.2 ± 2.8 (2.0–18.0)
	rBaux	92.4 ± 30.6 (23.0–201.0)
	Hangang	134.7 ± 22.5 (94.0–214.0)
	APACHE IV	50.9 ± 31.1 (0.0–167.0)
	SOFA	4.2 ± 3.4 (0.0–19.0)
Complications	Sepsis	1706 (65.4%)
	ARDS	1013 (38.8%)
	AKI	1430 (54.8%)
Interventions	Ventilator	1356 (52.0%)
	CRRT	375 (14.4%)

^1^ n (%); mean ± SD (range).

**Table 2 ijms-25-06769-t002:** Overview of Demographic Characteristics by Cluster.

Group	Characteristic	A, N = 461 ^1^	B, N = 453 ^1^	C, N = 730 ^1^	D, N = 964 ^1^	*p*-Value ^2^
Outcomes	Mortality	224 (48.6%)	54 (11.9%)	249 (34.1%)	114 (11.8%)	<0.001
	Length of Stay	21.5 ± 8.1 (6.0–30.0)	23.6 ± 7.3 (7.0–30.0)	8.5 ± 9.0 (1.0–30.0)	11.6 ± 9.7 (2.0–30.0)	<0.001
	AKI onset	7.8 ± 7.7 (1.0–30.0)	4.8 ± 6.5 (1.0–30.0)	3.1 ± 4.2 (1.0–30.0)	3.8 ± 5.6 (1.0–30.0)	<0.001
Patients’ Factors	Age	55.7 ± 15.9 (18.0–99.0)	52.0 ± 16.8 (18.0–95.0)	53.0 ± 15.7 (18.0–99.0)	50.6 ± 16.0 (19.0–96.0)	<0.001
	Sex					0.047
	Male	370 (80.3%)	378 (83.4%)	576 (78.9%)	743 (77.1%)	
	Female	91 (19.7%)	75 (16.6%)	154 (21.1%)	221 (22.9%)	
	TBSA	44.8 ± 26.2 (0.0–99.0)	35.4 ± 18.0 (0.0–85.0)	34.8 ± 29.3 (0.0–100.0)	26.4 ± 20.3 (0.0–100.0)	<0.001
	Inhalation	211 (45.8%)	167 (36.9%)	298 (40.8%)	306 (31.7%)	<0.001
	Type					<0.001
	FB	359 (77.9%)	328 (72.4%)	514 (70.4%)	617 (64.0%)	
	SB	47 (10.2%)	52 (11.5%)	57 (7.8%)	101 (10.5%)	
	EB	30 (6.5%)	39 (8.6%)	107 (14.7%)	175 (18.2%)	
	ChB	4 (0.9%)	7 (1.5%)	17 (2.3%)	18 (1.9%)	
	CoB	21 (4.6%)	27 (6.0%)	35 (4.8%)	53 (5.5%)	
Severity Scores	ABSI	9.7 ± 2.7 (4.0–16.0)	8.4 ± 1.9 (4.0–13.0)	8.4 ± 3.4 (3.0–18.0)	7.4 ± 2.4 (2.0–17.0)	<0.001
	rBaux	108.3 ± 26.7 (42.0–181.0)	93.6 ± 21.4 (41.0–154.0)	94.7 ± 37.0 (29.0–201.0)	82.3 ± 26.6 (23.0–197.0)	<0.001
	Hangang	141.2 ± 20.0 (97.0–201.0)	134.1 ± 15.6 (100.0–195.0)	137.7 ± 29.6 (94.0–214.0)	129.5 ± 18.6 (94.0–207.0)	<0.001
	APACHE IV	60.0 ± 26.9 (4.0–141.0)	50.0 ± 27.6 (5.0–148.0)	55.2 ± 37.0 (1.0–167.0)	43.8 ± 27.9 (0.0–152.0)	<0.001
	SOFA	5.3 ± 3.2 (0.0–18.0)	4.0 ± 2.9 (0.0–18.0)	4.6 ± 4.0 (0.0–19.0)	3.3 ± 2.9 (0.0–16.0)	<0.001
Complications	Sepsis	434 (94.1%)	405 (89.4%)	318 (43.6%)	549 (57.0%)	<0.001
	ARDS	335 (72.7%)	236 (52.1%)	201 (27.5%)	241 (25.0%)	<0.001
	AKI	385 (83.5%)	263 (58.1%)	377 (51.6%)	405 (42.0%)	<0.001
Interventions	Ventilator	382 (82.9%)	282 (62.3%)	337 (46.2%)	355 (36.8%)	<0.001
	CRRT	156 (33.8%)	52 (11.5%)	90 (12.3%)	77 (8.0%)	<0.001

^1^ n (%); Mean ± SD (Range). ^2^ Pearson’s Chi-squared test; One-way ANOVA.

**Table 3 ijms-25-06769-t003:** Diagnostic Performance and Predictive Validity of Biomarkers for AKI Across Patient Clusters.

Clusters	Predictors	AUC (95% CI)	Cutoff	Accuracy	Sensitivity	Specificity	PPV	NPV
Cluster A								
	PT	0.918 (0.886~0.949)	12.5	0.815	0.785	0.926	0.457	0.025
	Cystatin C	0.917 (0.882~0.952)	0.8	0.843	0.845	0.838	0.296	0.077
	eGFR cys	0.908 (0.868~0.947)	160	0.863	0.910	0.750	0.224	0.103
	Albumin	0.892 (0.859~0.925)	1.9	0.725	0.679	0.961	0.628	0.011
	TB	0.887 (0.851~0.923)	0.4	0.773	0.741	0.907	0.547	0.029
	RDW	0.886 (0.853~0.919)	12.8	0.781	0.764	0.868	0.580	0.033
	Creatinine	0.885 (0.852~0.919)	0.8	0.788	0.766	0.895	0.564	0.027
Cluster B								
	Albumin	0.875 (0.848~0.901)	2.6	0.766	0.741	0.800	0.306	0.165
	Creatinine	0.874 (0.848~0.900)	0.79	0.693	0.656	0.742	0.387	0.224
	PT	0.871 (0.840~0.902)	13.8	0.729	0.789	0.676	0.219	0.314
	WBC	0.848 (0.820~0.877)	5.98	0.719	0.779	0.637	0.324	0.253
Cluster C								
	Albumin	0.840 (0.804~0.876)	2.6	0.824	0.789	0.862	0.207	0.141
	PT	0.804 (0.759~0.849)	13.9	0.828	0.771	0.875	0.176	0.165
	Cystatin C	0.791 (0.738~0.845)	0.6	0.831	0.801	0.849	0.127	0.233
	eGFR cys	0.780 (0.723~0.836)	134	0.827	0.782	0.856	0.141	0.223
	Creatinine	0.764 (0.721~0.807)	1.11	0.821	0.808	0.836	0.196	0.161
	pH	0.764 (0.720~0.808)	7.358	0.814	0.771	0.861	0.228	0.140
	Lactate	0.762 (0.716~0.808)	0.8	0.816	0.769	0.862	0.208	0.154
	Platelet	0.757 (0.713~0.801)	265	0.802	0.724	0.886	0.250	0.128
	eGFR cr	0.757 (0.696~0.818)	85	0.811	0.792	0.832	0.222	0.157
	RDW	0.753 (0.708~0.797)	14.8	0.798	0.769	0.830	0.230	0.171
	TB	0.751 (0.705~0.797)	0.5	0.794	0.748	0.840	0.227	0.179
	BUN	0.749 (0.704~0.794)	34.2	0.797	0.762	0.836	0.233	0.169
Cluster D								
	Albumin	0.798 (0.768~0.828)	3.2	0.744	0.748	0.741	0.195	0.328
	PT	0.757 (0.720~0.793)	14.3	0.761	0.564	0.865	0.211	0.312
	Lactate	0.744 (0.709~0.779)	0.7	0.684	0.678	0.687	0.223	0.430
	Cystatin C	0.731 (0.686~0.776)	0.9	0.725	0.571	0.780	0.166	0.516
	Platelet	0.729 (0.695~0.762)	407	0.696	0.614	0.755	0.269	0.357
	RDW	0.723 (0.690~0.757)	14.6	0.706	0.536	0.828	0.288	0.308
	TB	0.721 (0.686~0.756)	1.6	0.713	0.527	0.832	0.267	0.333
	eGFR cys	0.721 (0.674~0.768)	160	0.665	0.649	0.671	0.162	0.578
	Creatinine	0.720 (0.686~0.754)	0.46	0.705	0.569	0.801	0.276	0.329
	WBC	0.719 (0.686~0.753)	11.04	0.710	0.448	0.898	0.307	0.241

## Data Availability

The datasets generated and/or analyzed during this study are available from the corresponding author upon reasonable request.

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
