# Peer review of "Deciphering AKI in Burn Patients: Correlations between Clinical Clusters and Biomarkers"

_ijms, 2024, doi:10.3390/ijms25126769_

Round 1

Reviewer 1 Report

Comments and Suggestions for Authors

This manuscript presents a comprehensive retrospective cohort study conducted at Hangang Sacred Heart Hospital's Burn Intensive Care Unit, focusing on the incidence and predictors of Acute Kidney Injury among burn patients.

1. Were the biomarker measured from the blood samples?

2. While Cystatin C and eGFR cys are highlighted as consistent predictors, the discussion should also acknowledge that their predictive power might vary and that further validation is necessary.

3. The discussion does not fully integrate the findings from all clusters and fails to provide a holistic view of how these clusters interact or compare.

4. Although the discussion acknowledges the study's limitations, it does so briefly and does not delve deeply into how these limitations might impact the findings and their applicability. For examples, the focus on a single-center cohort limits the applicability of the results to other settings. The clusters show significant variation in mortality rates, length of stay, and other clinical parameters, suggesting heterogeneity within each cluster. This could mean that some subgroups may have overlapping characteristics, potentially leading to less precise clinical recommendations for individual patients. The patient cohort spans a ten-year period. During this time, clinical practices, treatment protocols, and medical technology have evolved significantly. These changes can introduce variability in patient management and outcomes, making it challenging to attribute results solely to patient characteristics or the interventions studied. Since AKI is an acute condition, creatinine measurements were taken every 2-3 days from admission to discharge, which could have missed some changes in the days that measurements were not taken.

5. Figure 2A bottom labeling was distorted.

Author Response

This manuscript presents a comprehensive retrospective cohort study conducted at Hangang Sacred Heart Hospital's Burn Intensive Care Unit, focusing on the incidence and predictors of Acute Kidney Injury among burn patients.

  1. Were the biomarker measured from the blood samples?
    Response: Yes, the biomarkers (Cystatin C, eGFR cys, Albumin, Prothrombin Time, Total Bilirubin, and others) were indeed measured from blood samples. We have clarified this in the Methods section. The following sentence has been added: "All biomarkers were measured from blood samples collected from patients at the time of ICU admission and during their stay."
  2. While Cystatin C and eGFR cys are highlighted as consistent predictors, the discussion should also acknowledge that their predictive power might vary and that further validation is necessary.
    Response: We agree that while Cystatin C and eGFR cys have shown consistent predictive power, further validation is indeed necessary. We have expanded the discussion to acknowledge this point and suggest future research directions for validation.
  3. The discussion does not fully integrate the findings from all clusters and fails to provide a holistic view of how these clusters interact or compare.
    Response: We have revised the discussion to provide a more integrated and holistic view of the findings from all clusters. We have added a comparative analysis of the clusters to highlight how they interact and compare.
  4. Although the discussion acknowledges the study's limitations, it does so briefly and does not delve deeply into how these limitations might impact the findings and their applicability. For examples, the focus on a single-center cohort limits the applicability of the results to other settings. The clusters show significant variation in mortality rates, length of stay, and other clinical parameters, suggesting heterogeneity within each cluster. This could mean that some subgroups may have overlapping characteristics, potentially leading to less precise clinical recommendations for individual patients. The patient cohort spans a ten-year period. During this time, clinical practices, treatment protocols, and medical technology have evolved significantly. These changes can introduce variability in patient management and outcomes, making it challenging to attribute results solely to patient characteristics or the interventions studied. Since AKI is an acute condition, creatinine measurements were taken every 2-3 days from admission to discharge, which could have missed some changes in the days that measurements were not taken.
    Response: We have expanded the discussion on the study's limitations, detailing how they might impact the findings and their applicability. We also addressed the variability introduced by changes in clinical practices and the potential impact of measurement intervals on capturing AKI events.
  5. Figure 2A bottom labeling was distorted.
    Response: We apologize for the labeling distortion in Figure 2A. This has been corrected in the revised manuscript.

Reviewer 2 Report

Comments and Suggestions for Authors

I read with interest the paper by Lee et al. regarding AKI in burn patients. I suggest some revisions to improve the general comprehension and generalizability of the manuscript.

- The title in the version differs from what is reported in the pdf file. Please reconcile.

- The abstract is somewhat discursive, without a specific presentation of the general results of the study. I suggest including the cluster presentation, the most prominent analysis results, and the statistical significance.

- The authors mentioned, "The method with the highest count of conditions showing substantial differentiation (based on a statistically determined threshold) was chosen as the most effective in capturing the heterogeneity in our study of Acute Kidney Injury in burn patients." Please include the adopted threshold.

- The type of CRRT (adopted anticoagulation, filters) and duration may influence the outcomes (consider 10.1016/j.burns.2024.02.028 and 10.1159/000528861). Please include all these available information and comment on it.

Comments on the Quality of English Language

Minor editing of English language required.

Author Response

I read with interest the paper by Lee et al. regarding AKI in burn patients. I suggest some revisions to improve the general comprehension and generalizability of the manuscript.

- The title in the version differs from what is reported in the pdf file. Please reconcile.
Response: We have reconciled the title differences between the versions. The title now consistently reads "Deciphering AKI in Burn Patients: Correlations between Clinical Clusters and Biomarkers."

- The abstract is somewhat discursive, without a specific presentation of the general results of the study. I suggest including the cluster presentation, the most prominent analysis results, and the statistical significance.
Response: We have revised the abstract to be more concise and to include the key results, including cluster presentations and statistical significance.

- The authors mentioned, "The method with the highest count of conditions showing substantial differentiation (based on a statistically determined threshold) was chosen as the most effective in capturing the heterogeneity in our study of Acute Kidney Injury in burn patients." Please include the adopted threshold.
Response: We have included the statistically determined threshold used for identifying substantial differentiation among conditions in the Methods section.

- The type of CRRT (adopted anticoagulation, filters) and duration may influence the outcomes (consider 10.1016/j.burns.2024.02.028 and 10.1159/000528861). Please include all these available information and comment on it.
Response: We have added details regarding the type of CRRT, anticoagulation methods, and fluid  used, as well as their potential influence on outcomes.